# A label indicating an old year of establishment improves evaluations of restaurants and shops serving traditional foods

Tomoki Maezawa *⊕, Jun I. Kawahara⊕

Department of Psychology, Hokkaido University, Sapporo, Hokkaido, Japan

⊕ These authors contributed equally to this work.
* shortfinned@gmail.com

## Abstract

This study examined whether the presence of product information focused on a past era (e.g., year of establishment) improved consumers' evaluations of a shop serving traditional products when the label and shop were congruent in terms of temporal focus. Across five experiments, participants viewed and evaluated advertisements from traditional food restaurants and shops that showed an old year of establishment. They showed favorable evaluations of the restaurants and food shops more frequently when a label focused on the past was displayed than when the label was absent or when a label focused on the present was displayed. Subsequent experiments indicated that this labeling effect was strongest when the label and shop were consistent in terms of traditional culture such that the year of establishment on the label showed the Japanese era name (Japan's traditional date format) and was accompanied by Japanese classic foods. Importantly, in this study, qualitative domains were consistently improved more often than were ratings of visit intention and expected taste. The results suggest that temporal congruence between the label and restaurants rated plays an essential role in ensuring that these advertisements are effective in improving positive evaluations.

## Introduction

The presence of additional information, such as a short sentence describing a product name or a brand, may help observers to evaluate the likeability or quality of a target product [1]. A wide variety of studies across psychological domains have demonstrated the effects of important symbolic features presented in visual images of commercial advertisements (for a review, see [1]). For example, labeling safety or nutrition information with a photograph of food products consistently alters impressions of the advertised products or brands [2–8]. Notably, marketing researchers [9] have reported misperception of symbolic terms that refer to a product characteristic (e.g., a "fruit" symbolizing healthiness), so that the symbolized meaning supersedes the product's negative characteristics (e.g., high-calorie content of fruit sugar). However, recent studies have demonstrated that positive influences of product labels enhance consumers' purchase intentions beyond the perceptual influences related to the mere presence of food labels [10–12].

**Data Availability Statement:** The data and stimuli are available via the Open Science Framework (https://osf.io/nf8sw/), and none of the experiments were preregistered.

**Funding:** This work was financially supported by a Grant-in-Aid from the Japan Society for the Promotion of Science Fellows (20J20490) to TM and a Grants-in-Aid for Scientific Research from the Japan Society for the Promotion of Science (20H01779) and (20H04568) to JK. The funders had no role in study design, data collection and analysis, decision to publish, or preparation of the manuscript.

**Competing interests:** The authors have declared that no competing interests exist.

An interaction between the two main components of advertisements, product images and labeling text, may play an important role in encouraging consumer purchase intentions [1]. For example, symbolic labels will evoke more favorable product attitudes when label-derived extrinsic expectations are consistent with the intrinsic perceptions of specific products (e.g., the label "organic" evokes a favorable attitude toward a yogurt product [13]). Similar interactions have been demonstrated via time-related perceptual congruence between labels and products [10–12,14]. Traditional, antique, historical, or classic products or brands are substantially preferred when the schemes are enhanced by temporal perception toward the past, rather than the present [10–12]. This conceptual congruence may facilitate information processing regarding these materials and indirectly influence consumers' shopping preferences [12,15].

Given this congruence effect, it seems reasonable that restaurateurs advertise their history to attract consumers. The presentation of traditional or historical symbols, such as the year of establishment, may aim to enhance consumer preference for the associated restaurant and increase consumer intentions to visit. For example, a large number of Japanese soba noodle (i.e., buckwheat-noodle) restaurants use this advertising strategy on their websites. Indeed, 77 of 100 soba noodle restaurant websites retrieved by preliminary online searches indicated their year of establishment on their main webpages. As shown in Fig 1, the years of establishment displayed on the webpages were distributed over a broad time range (mean year = 1916; median year = 1926; year range: 1465–2009). Because soba noodles are traditional and the historical Japanese diet arose before the medieval period, we hypothesized that temporal congruence effects would occur and elicit favorable consumer evaluations of the restaurants in the presence of past-focused labeling (i.e., an old year of establishment) compared with scenarios where the past-focused information was absent.

Importantly, this past-focused labeling itself can elicit favorable evaluations, consistent with the notion that a traditional description elicits positive consumer attitudes toward the food manufacturing processes [16]. Moreover, a focus on tradition in advertisements can enhance perceptions of quality (e.g., [17]). This tendency has been reported in the context of applied research demonstrating that elderly consumers prefer long-known products [18,19] and a

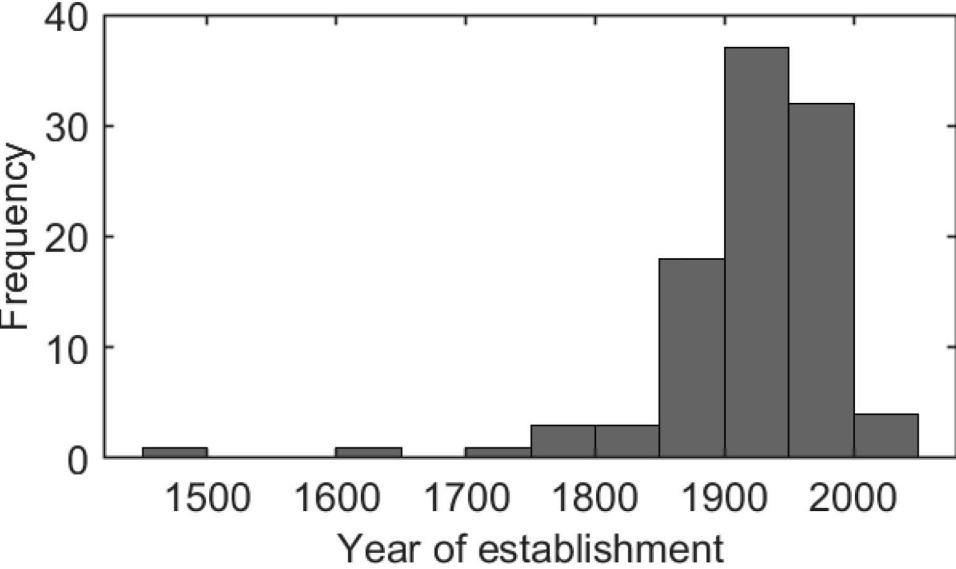

**Fig 1. Year of establishment data for soba noodle restaurants in Japan.** The data are from 100 soba noodle restaurant websites.

focus on authenticity [20]. Specifically, providing past-focused information, such as the year of establishment, promotes more favorable product appraisals. This effect is similar to the label-presentation effects seen for well-known brands [1,21] and traditional production processes [16] emphasizing historic aspects. Based on the idea that these associations are considered to indicate quality, we assumed that the presentation of the year of establishment in advertisements could improve consumer attitudes toward restaurants.

Altogether, consumer preference for restaurants can be enhanced by temporal congruence and past-focused labeling. Labeling effects on restaurant evaluations occur when the label emphasizes the historical nature of traditional foods. The present study examined whether temporal congruence improved evaluations of restaurants and shops. We measured three components of evaluations, similar to previous studies (e.g., [22]): product quality, expected taste, and visit intentions. Based on the notion that a traditional label symbolizes high quality [16,17], the presentation of a past-focused label should improve perceptions of a restaurant's quality. In Experiment 1 and 2, we compared ratings of advertisements of soba noodle restaurants that included an old year-of-establishment label with ratings of advertisements that lacked such a label. In Experiment 3, we examined whether temporal congruence effects occurred in ratings for advertisements for food shops other than soba noodle restaurants. We predicted that advertisements that included a past-focused label would more frequently evoke favorable evaluations of restaurants and shops with respect to product quality than would control advertisements that lacked such a label.

We measured participants' visit intentions to determine whether advertisements directly affected consumer behavior. The intention to consume foods or visit restaurants is distinct from attitudes regarding product quality and expected taste. According to the theory of planned behavior, attitudes predict intentions, and intentions predict behaviors [23,24]. According to this theory, attitudes regarding expected taste and product quality should be critical determinants of consumers' willingness to visit restaurants. A favorable attitude toward product quality may modulate taste expectations, resulting in favorable evaluations of food. Similarly, more favorable attitudes toward quality should directly or indirectly motivate consumers to visit restaurants. Thus, a secondary objective of the present study was to examine whether food quality evaluations directly or indirectly affect the intention to visit restaurants. We hypothesized that attitudes about quality modulate both taste expectations and visit intentions.

## Experiment 1

In Experiment 1, we examined whether participants evaluated soba-noodle restaurants that included a product label with an old year of establishment more favorably than those that lacked such a label, in terms of visit intention, expected taste, and product quality. Accordingly, we sequentially displayed two visual images of fictitious restaurant websites to participants. The year-of-establishment label was embedded in each restaurant's website image. A label indicating a long-past era may be considered to reflect high quality [16,17]. Based on this finding, we predicted that temporal congruence between the past-focused label and the image of soba noodles would improve impressions of the restaurant with respect to product quality compared with restaurants that lacked this congruence.

### Materials and methods

**Participants.** A group of 34 Japanese individuals (18 female participants; mean age = 45.74 years; range: 30–64 years) from a participant pool at an online crowdsourcing service for survey research was recruited and participated independently. The sample size was

**Table 1. Summary of experimental conditions.**

| Study | Participant's year range | Label condition | Lable type | Food type | Measure |
|---|---|---|---|---|---|
| Experiment 1 | 30–64 ($M = 45.74$) | absent vs. present | Japanese calendar | Soba noodles | quality, expected taste, and visit intention |
| Experiment 2 | 31–68 ($M = 49.53$) | modern vs. past | Japanese calendar | Soba noodles | quality, expected taste, and visit intention |
| Experiment 3 | 18–31 ($M = 19.90$) | absent vs. present | Gregorian calendar | cheese, hamburgers, wine, or Japanese confections | quality, expected taste, and visit intention |
| Experiment 4 | 23–74 ($M = 48.03$) | absent vs. present | Gregorian calendar | Japanese confections | quality, expected taste, and visit intention |
| Experiment 5 | 21–64 ($M = 45.88$) | absent vs. present | Japanese calendar | Japanese confections | quality, expected taste, and visit intention |

determined based on a power analysis performed using G*Power 3.1 software [25,26] with an effect size of 0.25 and power of 0.80. All participants provided written informed consent prior to the experiment and received a monetary payment. All experiments in this study were approved by the Human Research Ethics Committee of Hokkaido University.

**Stimuli.** Table 1 summarizes the experimental conditions used in the experiments. The experiment was performed using online presentations of visual images of fictitious restaurant websites (Fig 2). Pictures of two soba noodle restaurants (1,657 × 1,133 pixels) were used in Experiment 1. Each website included a main photograph of soba noodles, the name of the restaurant, and website navigation menus. A label showing the same year of establishment (80 × 80 pixels per Japanese character) appeared on the left side of the top picture on each website, vertically or horizontally. The label displayed the year established in Japan's Taisho 15 era date format (i.e., "大正15年"), which corresponded to the median year (= 1926 C.E.) of 100 soba-noodle restaurants retrieved from our prior web searches.

**Procedure.** Fig 2 depicts an example of a trial sequence in Experiment 1. The two restaurant website pictures were presented sequentially on the web during online surveys. One of the

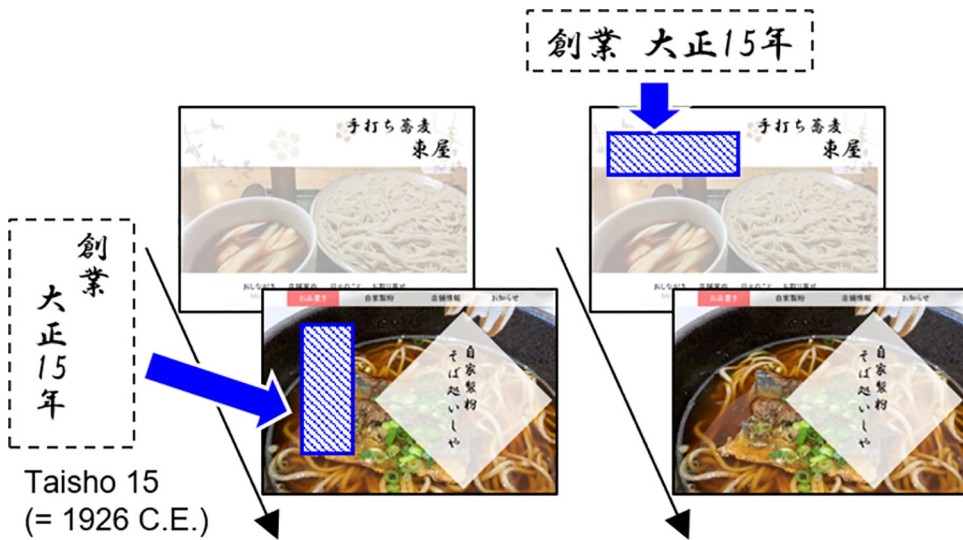

**Fig 2. Schematic of a trial sequence from Experiment 1.** The restaurant's year of establishment was presented in a label in Japan's Taisho 15 era date format (1926 C.E.). The label condition (present vs. absent) was counterbalanced across participants.

pictures included a year label, but the other did not. The order of label presentation was counter-balanced across participants. For example, a group of participants sequentially observed the picture of Restaurant A in the presence of a year label and Restaurant B in the absence of the label. The opposite group viewed Restaurant A in the absence of the label and Restaurant B in the presence of the label. Participants were instructed to view each picture for 30 seconds and memorize it. After a single presentation of each stimulus, participants indicated their attitudes toward the restaurant in terms of product quality and expected taste, and their intention to visit, via the questions "How good is the quality?", "How good is the taste?", and "How badly do you want to visit?", respectively. Responses were made via a seven-point scale (1 = low visit intention, bad taste, bad quality; 7 = high visit intention, good taste, good quality). Participants then reported the name of the food (i.e., soba noodles) that was presented on the website to test compliance with the task instructions. The data obtained from the online survey were analyzed using a repeated-measures multivariate analysis of variance (MANOVA) with the label condition (label presence vs. label absence) as the within-subjects factor, using SPSS software (IBM Corp.).

## Results and discussion

All participants correctly identified the name of the food referred to on the restaurants' websites (i.e., soba noodles). The mean scores of participants' evaluations were calculated separately for the three dependent variables: visit intention, expected taste, and product quality (Fig 3). The repeated-measures MANOVA demonstrated a significant main effect of label condition on the ratings [$F(3, 31) = 7.47$, $p = .001$, Wilks' $\lambda = .581$, $\eta_p^2 = .419$], indicating that evaluations of the restaurant improved in the presence of the year label. Follow-up univariate tests with Bonferroni corrections revealed significant effects of the label condition on ratings for visit intention [$F(1, 33) = 9.76$, $p = .012$, $\eta_p^2 = .228$], expected taste [$F(1, 33) = 8.13$, $p = .021$, $\eta_p^2 = .198$], and product quality [$F(1, 33) = 21.37$, $p < .001$, $\eta_p^2 = .393$].

We assessed whether attitudes regarding quality and expected taste correlated with the intention to visit using Model 4 of Hayes' PROCESS SPSS macros [27]. A mediation analysis was performed (where 0 = label absent and 1 = label present) including ratings of expected taste as a mediator. Quality rating was an independent variable and the intention to visit rating was the dependent variable (5,000 bootstrap samples). Table 2 summarizes the results of the mediation analysis. The results revealed a significant effect of product quality ratings on the

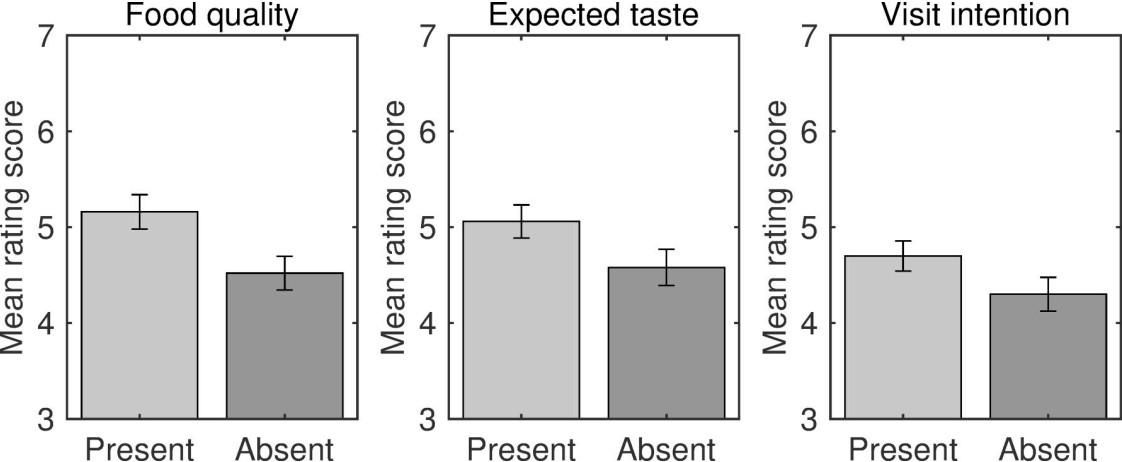

**Fig 3. Evaluations of restaurants in Experiment 1.** Bar plots for visit intention, expected taste, and food quality ratings, are shown separately for the label present and label absent conditions. Error bars indicate standard error.

**Table 2. Summary of the MANOVA and mediation analysis result.**

| Study | Label effect ($\eta p2$) | Direct effect ($d$) | Indirect effect ($d$) |
|---|---|---|---|
| **Experiment 1** | .419** | | |
| label-absent | | .21 | .51** |
| label-present | | .04 | .40** |
| **Experiment 2** | .321** | | |
| present-focused label | | .47** | .33 |
| past-focused label | | .20 | .54** |
| **Experiment 3** | | | |
| cheese | .366** | | |
| label-absent | | .33 | .26** |
| label-present | | .25 | .31** |
| hamburgers | .284** | | |
| label-absent | | .13 | .44** |
| label-present | | .31 | .29** |
| wine | .231** | | |
| label-absent | | .34 | .23** |
| label-present | | .33 | .28** |
| Japanese confections | .180 | | |
| label-absent | | .34 | .19 |
| label-present | | .36 | .28 |
| **Experiment 4** | .032 | | |
| label-absent | | .62** | .21 |
| label-present | | .28 | .53** |
| **Experiment 5** | .324** | | |
| label-absent | | -.06 | .51** |
| label-present | | -.03 | .82** |

**significant result. The MANOVA revealed significant effects of labeling on restaurant evaluations, while the mediation analysis showed significant direct and/or indirect effects of attitudes about quality on visit intentions.

intention to visit ratings under the label-absent [$b = .71$, $SE = 0.13$, $t = 5.32$, $p < .001$] and label-present conditions [$b = .45$, $SE = 0.14$, $t = 3.22$, $p = .003$]. However, when the expectation of food taste was included as a mediator, significance disappeared in both the label-absent [$b = .21$, $SE = 0.22$, $t = 0.95$, $p = .348$] and label-present conditions [$b = .04$, $SE = 0.21$, $t = 0.22$, $p = .829$]. The indirect effect of quality ratings on intention ratings was significant when the label was absent [$b = .51$, $SE = 0.26$, 95% CI [0.041, 1.034]] and when it was present [$b = .40$, $SE = 0.17$, 95% CI [0.085, 0.779]].

The results of Experiment 1 demonstrated that the presence of a label indicating an old year of establishment improved participants' evaluations of the restaurant. As expected, the results revealed that the effect size of the quality rating ($\eta_p^2 = .393$) was higher than that of visit intention ($\eta_p^2 = .228$) and expected taste ($\eta_p^2 = .198$). This result showing high quality evaluations indirectly indicated that the traditional label symbolized quality, similar to the notion that past-focused information symbolizes quality and authenticity [17,20]. This finding implies that the old year-of-establishment label may induce participants' time perception to move in the past direction in a manner congruent with the temporal focus of traditional restaurants. We presume that this temporal congruence between the label and product perception played an essential role in that participants' evaluations of the restaurant improved with respect to product quality, as indicated in Experiment 1.

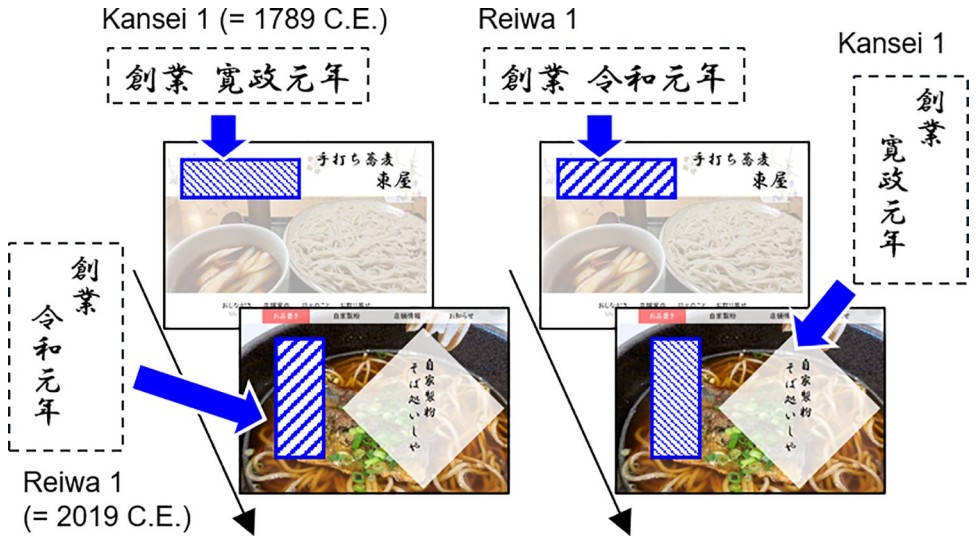

**Fig 4. Schematic of a trial sequence in Experiment 2.** The restaurant's year of establishment was presented as the label displayed in Kansei 1 (= 1789 C.E.) and Reiwa 1 (= 2019 C.E.) using Japanese traditional era names. The past-focused and present-focused label conditions were counterbalanced across participants. Food pictures are similar, but not identical, to the original images and are therefore for illustrative purposes only.

The results of a regression analysis supported the mediation model, where taste expectations modulated the relationship between attitudes about product quality and visit intentions. No direct relationship was found between attitudes about quality and intention to visit. Moreover, the label condition did not affect the observed indirect effect of the attitude toward quality. These findings suggest that attitude toward quality was positively correlated with taste expectations, independent of the presence or absence of the label, and thus indirectly increased consumers' willingness to visit the restaurant due to favorable taste expectations.

However, a concern in the results of Experiment 1 was that an information-rich advertisement (i.e., a restaurant webpage with the year-of-establishment label) was preferred relative to

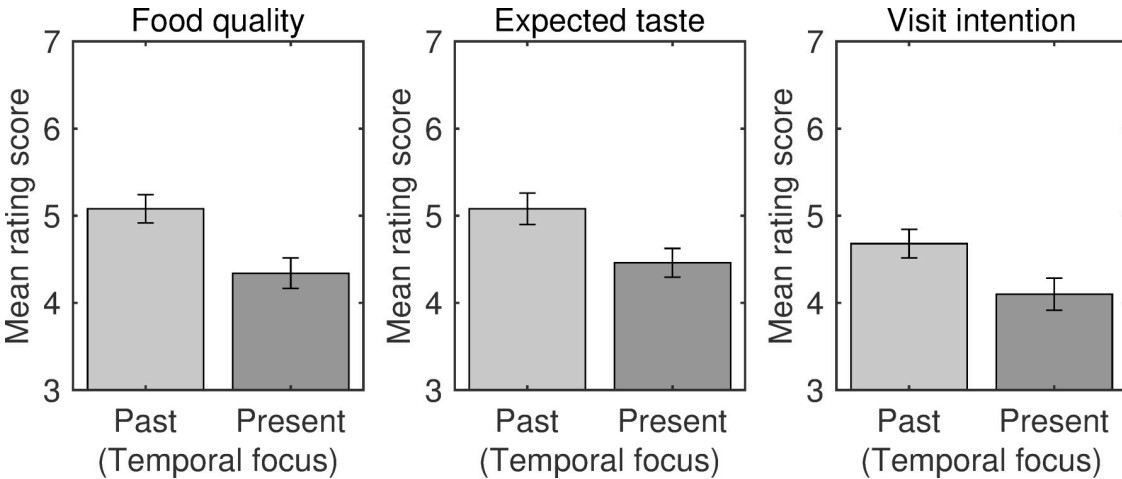

**Fig 5. Evaluations of restaurants in Experiment 2.** Bar plots for visit intention, expected taste, and food quality ratings are shown separately for the past-focused and present-focused label conditions. Error bars indicate standard error.

its less informative counterpart (i.e., a webpage without such a label). Although unlikely, this confounding factor required attention before we could clearly conclude that a label indicating year of establishment elicited past time-related perception. If the temporal direction emphasized by the year label played a critical role in the pattern of results in Experiment 1, older years should elicit stronger positive evaluations. Alternatively, if the presence of the label alone were sufficient to induce the enhanced evaluations indicated in Experiment 1, the temporal focusing direction symbolized by the label would not be determinative. To rule out this alternative explanation of the effects of mere label presentation, Experiment 2 compared restaurant evaluations between two types of year labels focusing on past and present periods.

## Experiment 2

In Experiment 2, we compared evaluations of restaurants in the presence of a past-focused label with evaluations of restaurants in the presence of a present-focused label, to rule out the possibility that the label presentation itself caused the effects observed in Experiment 1. To achieve this comparison, two visual images of fictitious restaurant websites were sequentially displayed to participants using the same procedure as in Experiment 1. These website pictures included two versions of the time-related information: one with a past-focused label and the other with a present-focused label, with the year of establishment depicted using the Japanese era name. We predicted that presentation of the past-focused label would emphasize the qualitative aspects of restaurants more than a present-focused label would.

### Materials and methods

A new group of 34 participants (10 female participants; mean age = 49.53 years; age range = 31–68 years) was recruited to Experiment 2. The stimuli and procedure (Fig 4) were identical to those used in Experiment 1, except that the year-of-establishment labeling was past-focused, i.e., "寛政元年" (Japan's Kansei 1 era date format, = 1789 C.E.) or present-focused, i.e., "令和元年" (Japan's Reiwa 1 era date format, = 2019 C.E.). The pictures of restaurant websites were presented in a web-based format during online surveys. The order of the two types of label presentations was counterbalanced across participants. For example, one group of participants sequentially observed the picture of Restaurant A with a past-focused label and Restaurant B with a present-focused label. The other group observed Restaurant A with a present-focused label and Restaurant B with a past-focused label. Participants indicated their three evaluations of the restaurants in the same manner as in Experiment 1, i.e., visit intention, expected taste, and product quality, on a seven-point scale. At the end of the survey, participants reported the name of the food (i.e., soba noodles) that was presented on the websites to test compliance with the task instructions. The data were analyzed using a repeated-measures MANOVA with temporal focus (past vs. present) as the within-subjects factor.

### Results and discussion

All participants in Experiment 2 correctly identified the food name (i.e., soba noodles) referenced in the restaurant websites. The mean scores of participants' evaluations were calculated separately for the dependent valuables (visit intention, expected taste, and product quality; Fig 5). A repeated-measures MANOVA revealed a significant main effect of temporal focus on the ratings [$F(3, 31) = 4.87$, $p = .007$, Wilks' $\lambda = .679$, $\eta_p^2 = .321$]. Follow-up univariate tests with Bonferroni corrections revealed significant effects of temporal focus on ratings for visit intention [$F(1, 33) = 11.94$, $p = .006$, $\eta_p^2 = .266$], expected taste [$F(1, 33) = 13.10$, $p = .003$, $\eta_p^2 = .284$], and product quality [$F(1, 33) = 13.67$, $p = .003$, $\eta_p^2 = .293$].

Mediation analyses were performed separately for the two label conditions (where 0 = present-focused label and 1 = past-focused label; 5,000 bootstrap samples). The results revealed a significant effect of product quality ratings on visit intention ratings under the present-focused [$b$ = .80, $SE$ = 0.10, $t$ = 7.80, $p$ < .001] and past-focused label conditions [$b$ = .74, $SE$ = 0.14, $t$ = 5.27, $p$ < .001]. When expectation of food taste mediated the correlation between quality and intention ratings, a significant direct effect was observed under the present-focused label condition [$b$ = .47, $SE$ = 0.13, $t$ = 3.52, $p$ = .001], whereas the direct effect was not significant under the past-focused label condition [$b$ = .20, $SE$ = 0.17, $t$ = 1.20, $p$ = .239]. The indirect effect of quality ratings on visit intention ratings was not significant under the present-focused label condition [$b$ = .33, $SE$ = 0.14, 95% CI [−0.044, 0.536]], whereas a significant indirect effect was observed under the past-focused-label condition [$b$ = .54, $SE$ = 0.17, 95% CI [0.238, 0. 898]].

The results of Experiment 2 demonstrated that participants' evaluations of restaurants were improved when a past-focused label was used, compared with the use of a present-focused label. These results ruled out the alternative explanation that label presentation referring to a particular temporal period alone contributed to the positive evaluations observed in Experiment 1. Instead, the direction of temporal focus appeared to influence restaurant evaluations. The results also revealed trends analogous to those of Experiment 1 in that the effect size of quality rating ($\eta_p^2$ = .293) was greater than the effect sizes of visit intention ($\eta_p^2$ = .266) and expected taste ($\eta_p^2$ = .284). Thus, the results of Experiment 2 suggest that the congruence of temporal focus between the label and product perception (i.e., toward a past era) positively modulated evaluations of the restaurants with respect to these qualitative aspects.

Moreover, the mediation analysis results differed by the label condition. When the label showed past-focused information, the results were consistent with Experiment 1, i.e., food taste expectations significantly mediated the correlation between the quality ratings and the intention to visit the restaurants. By contrast, when the label showed present-focused information, favorable attitudes toward product quality increased visit intentions directly. Although an indirect effect was not always present, we suggest that improving attitudes toward quality is likely to increase consumers' willingness to visit restaurants both directly and indirectly. Experiments 1 and 2 used different labeling conditions (i.e., no label condition in Experiment 1 and present-focused label condition in Experiment 2). The inconsistencies in the results may be due to the differences in labeling conditions.

The findings of Experiments 1 and 2 support the notion that a traditional advertising strategy can improve consumers' evaluations of a restaurant. However, the results of Experiments 1 and 2 were limited to effects in response to soba noodle restaurants and the participants largely included middle-aged or elderly populations (mean age = 47.63 years for participants in Experiments 1 and 2) rather than younger populations. We could not rule out the possibility that the mostly older participants in Experiments 1 and 2 tended to prefer products with old years of establishment due to an age-related bias (e.g., [18,19]). Accordingly, the results of Experiments 1 and 2 might have been primarily associated with this aspect of age-related brand loyalty rather than a preference induced by temporal congruence [10–12] that may not depend on consumer age. To exclude this possibility, Experiment 3 aimed to replicate the temporal congruence effects found in Experiment 1 using populations younger than those who had participated in Experiments 1 and 2.

Experiment 3 aimed to generalize the year-labeling effect to popular foods in Japan (i.e., other than soba noodles). Prior to this experiment, we conducted a preliminary survey to select food shop stimuli in which the year of establishment was advertised on main webpages or product packages. Based on the results of that survey, Experiment 3 used four types of popular foods in Japan: cheese, hamburgers, wine, and Japanese confections (Wagashi, a classic Japanese dessert). Because these types of foods were developed in the remote past, a label focused

on a past direction would highlight the qualitative aspects of these food shops. Thus, Experiment 3 aimed to replicate the temporal congruence effects observed in Experiment 1.

## Experiment 3

In Experiment 3, we examined whether participants had more favorable evaluations of food shops with an old year-of-establishment label compared with those that lacked such a label. To generalize the temporal congruence effects, we presented pictures of shops serving popular foods in Japan (other than soba noodles) to participants. The experimental procedure was nearly identical to that used in Experiment 1. To reduce the influence of age-related preferences (e.g., [18,19]) on the present conclusions, Experiment 3 recruited participants younger than those who had participated in Experiments 1 and 2. Based on the notion that a traditional label symbolizes high quality [16,17], the participants were expected to have favorable evaluations in terms of product quality toward restaurants with an old year-of-establishment label compared with restaurants that lacked such a label.

### Materials and methods

A new group of 136 participants (57 female participants; mean age = 19.9 years; age range: 18–31 years) was assigned randomly to rate four foods (i.e., cheese, hamburgers, wine, and Japanese confections). Each group comprised 34 participants. All participants were recruited from a participant pool at Hokkaido university in Japan and provided written informed consent prior to participation in the experiment.

The experiment was performed in a university laboratory through visual presentation of two fictitious pictures of food shop websites (Fig 6) on an LCD monitor (100 Hz refresh rate, 1,920 × 1,080 pixels, XL2411T; BenQ). Each picture (1,501 × 1,126 pixels) consisted of a main photograph of each food, the name of the shop, and website navigation menus. On each website, a year-of-establishment label (80 × 80 pixels per Japanese character) was horizontally embedded in the space at the top of the main photograph. The year of establishment was displayed as the Gregorian calendar year "1912" for all types of food shops.

A similar procedure to that used in Experiments 1 and 2 was employed so that participants sequentially observed two pictures, one with the label and the other without the label. In Experiment 3, the labeled shops were presented before or after the unlabeled shops using an order randomly determined for individual participants. For example, one group of the participants observed a labeled picture of Restaurant A before (or after) they observed an unlabeled picture of Restaurant B, and the other group observed an unlabeled picture of Restaurant A before (or after) they observed a labeled picture of Restaurant B (Fig 6). Participants were instructed to view each picture for 30 seconds and memorize the stimuli. After a single presentation of each stimulus, participants indicated their evaluations of the shop in terms of visit intention, expected taste, and product quality, using a seven-point scale. At the end of the survey, participants reported food names presented on the screen to test their compliance with the task instructions. To comprehensively analyze the influence of food type on the labeling effects, the data were subjected to a repeated-measures MANOVA with the label condition (label present vs. label absent) as the within-subjects factor and food types (cheese, hamburgers, wine, and Japanese confections) as the between-subjects factor.

### Results and discussion

All participants in Experiment 3 correctly identified the food types from the shops' website pictures. The mean rating scores were calculated separately for the dependent values (visit intention, expected taste, and product quality) and food types (Fig 7).

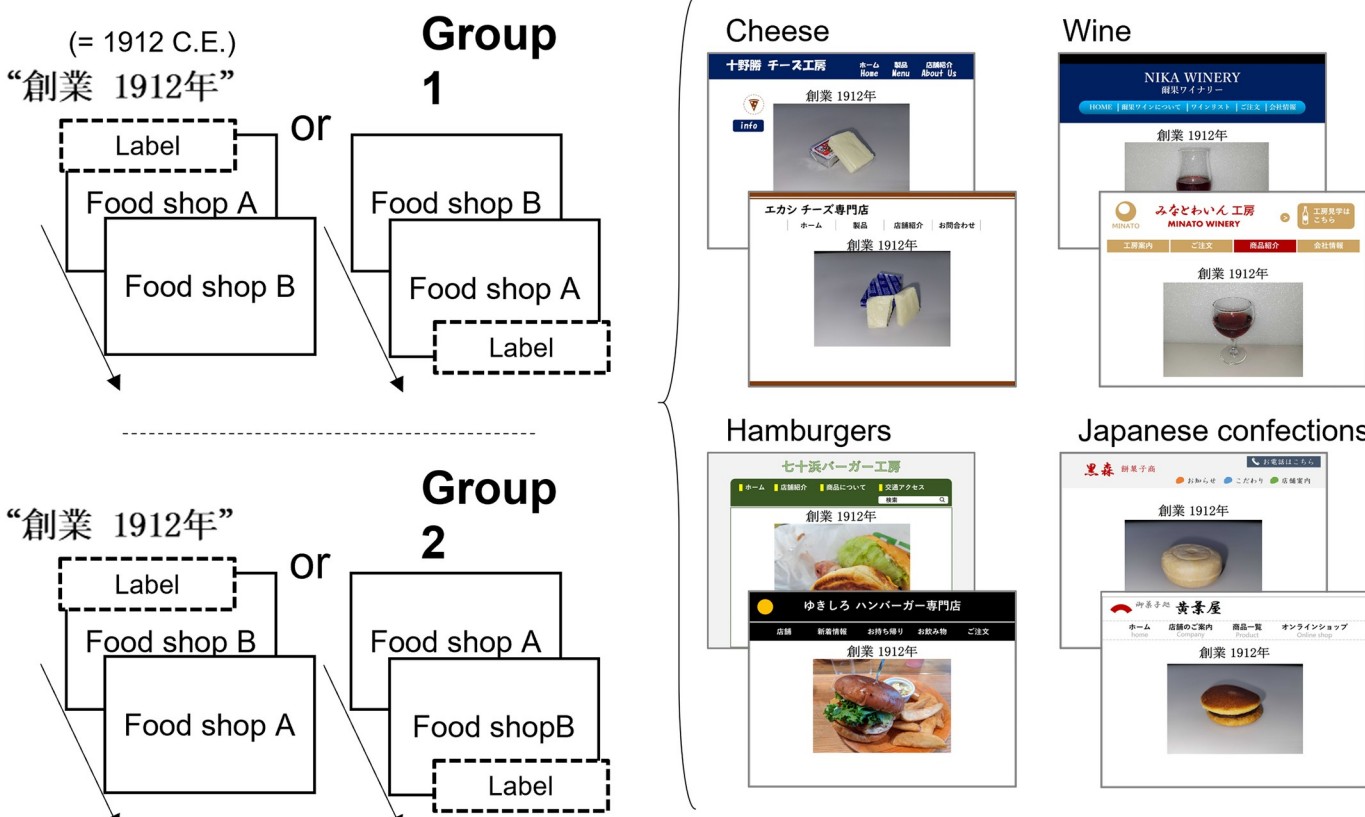

**Fig 6. Schematic of a trial sequence from Experiment 3.** The label-present and -absent conditions were counterbalanced across participants. A group of participants randomly observed a labeled picture before (or after) observing an unlabeled picture.

Repeated-measures MANOVAs were performed separately for the different food types. For cheese, a significant main effect of label condition on the ratings was found [$F(3, 31) = 5.96$, $p = .002$, Wilks' $\lambda = .634$, $\eta_p^2 = .366$]. Follow-up univariate tests with Bonferroni corrections revealed a significant effect of label condition on product quality ratings [$F(1, 33) = 17.04$, $p < .001$, $\eta_p^2 = .341$], whereas the effects on ratings of visit intention [$F(1, 33) = 1.04$, $p = .942$, $\eta_p^2 = .031$] and expected taste [$F(1, 33) = 0.17$, $p = 1.00$, $\eta_p^2 = .005$] were not significant. For hamburgers, MANOVA revealed a significant main effect of label condition [$F(3, 31) = 4.10$, $p = .015$, Wilks' $\lambda = .716$, $\eta_p^2 = .284$]. Univariate tests indicated that the main effect of label condition on quality ratings was not significant [$F(1, 33) = 4.40$, $p = .132$, $\eta_p^2 = .118$], whereas there was a significant effect on ratings of visit intentions [$F(1, 33) = 12.44$, $p = .003$, $\eta_p^2 = .274$] and expected taste [$F(1, 33) = 8.09$, $p = .024$, $\eta_p^2 = .197$]. For wine, MANOVA revealed a significant main effect of label condition [$F(3, 31) = 3.11$, $p = .041$, Wilks' $\lambda = .769$, $\eta_p^2 = .231$]. However, univariate tests revealed no significant main effect of label condition on quality ratings [$F(1, 33) = 3.51$, $p = .210$, $\eta_p^2 = .096$], visit intention [$F(1, 33) = 1.51$, $p = .681$, $\eta_p^2 = .044$], or expected taste [$F(1, 33) = 0.405$, $p = 1.00$, $\eta_p^2 = .012$]. For Japanese confections, MANOVA revealed no significant main effect of label condition [$F(3, 31) = 2.26$, $p = .101$, Wilks' $\lambda = .820$, $\eta_p^2 = .180$]. Taken together, the results showed a labeling effect for all three Western foods, but not for the traditional Japanese food. The labeling effect for Japanese confections was inconsistent with that observed in Experiments 1 and 2. The difference may be due the use of a Japanese calendar (Experiments 1 and 2) versus the Gregorian calendar (Experiment 3).

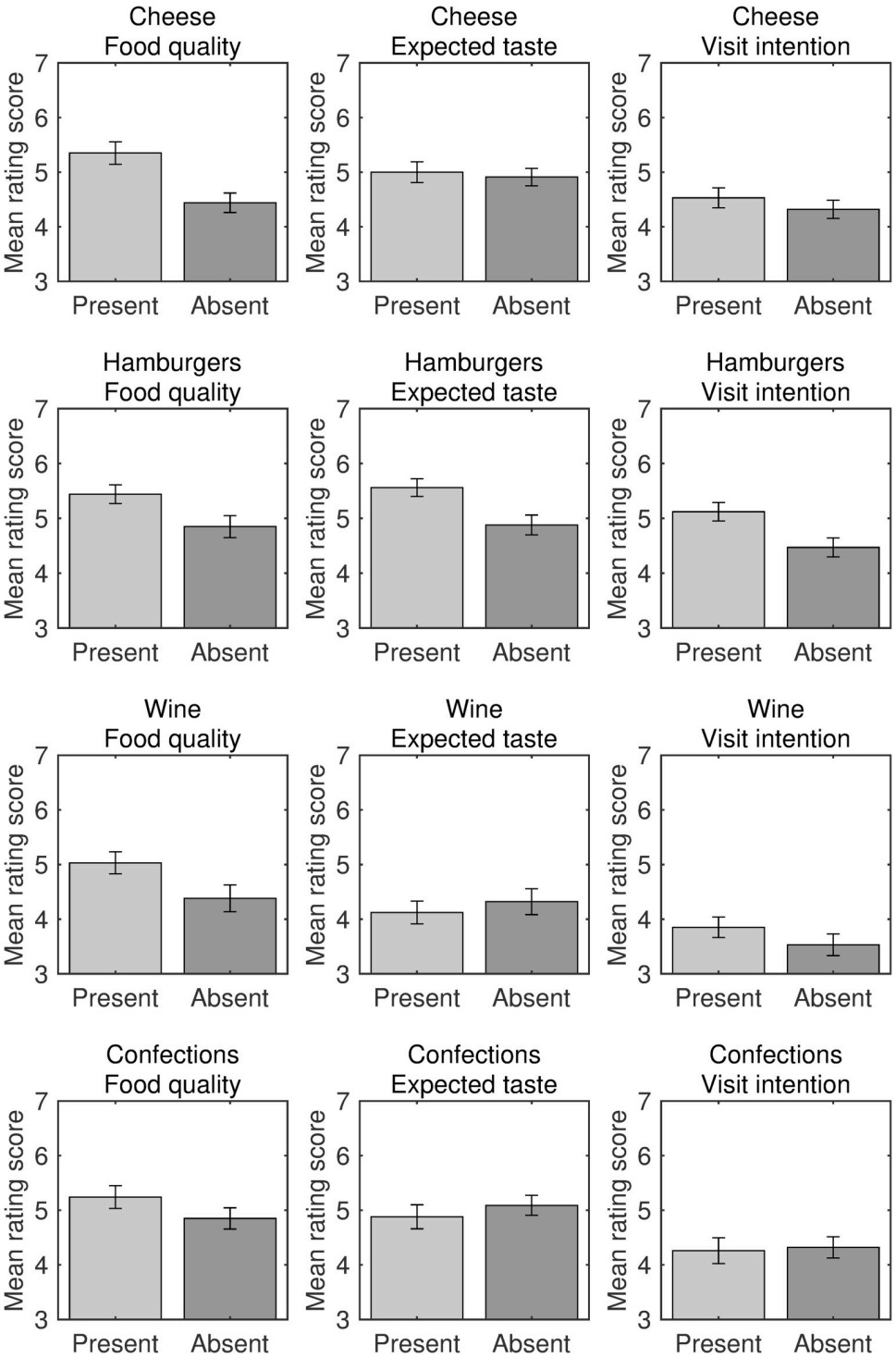

**Fig 7. Evaluations of food shops in Experiment 3.** Bar plots for visit intention, expected taste, and food quality ratings, and types of foods (i.e., cheese, hamburgers, wine, and Japanese confections), are shown separately for the label present and label absent conditions. Error bars indicate standard error.

To examine the influence of food types on the labeling effects, a repeated-measures MANOVA was performed with food type as the between-subjects factor. The results revealed significant main effects of label condition [$F(3, 130) = 9.42$, $p < .001$, Wilks' $\lambda = .821$, $\eta_p^2 = .179$] and food type [$F(9, 316.5) = 3.01$, $p = .002$, Wilks' $\lambda = .819$, $\eta_p^2 = .064$] on the ratings. However, the two-way interaction term was not statistically significant [$F(9, 316.5) = 1.77$, $p = .074$, Wilks' $\lambda = .887$, $\eta_p^2 = .039$], indicating that food type did not modulate the effect of label presentation. Follow-up univariate analyses with Bonferroni correction revealed a significant effect of label condition on product quality rating [$F(1, 132) = 20.94$, $p < .001$, $\eta_p^2 = .137$], but the effects on ratings of visit intention [$F(1, 132) = 5.68$, $p = .057$, $\eta_p^2 = .041$] and expected taste [$F(1, 132) = 0.48$, $p = 1.00$, $\eta_p^2 = .004$] were not statistically significant.

We also performed separate mediation analyses for the food type and label conditions. The results showed significant effects of product quality ratings when the label was absent [$b = .59$, $SE = 0.13$, $t = 4.96$, $p < .001$ for cheese; $b = .56$, $SE = 0.11$, $t = 4.96$, $p < .001$ for hamburger; $b = .56$, $SE = 0.10$, $t = 5.53$, $p < .001$ for wine; $b = .52$, $SE = 0.15$, $t = 3.51$, $p = .001$ for Japanese confections] and when it was present [$b = .55$, $SE = 0.12$, $t = 4.55$, $p < .001$ for cheese; $b = .60$, $SE = 0.14$, $t = 4.35$, $p < .001$ for hamburger; $b = .61$, $SE = 0.12$, $t = 4.95$, $p < .001$ for wine; $b = .64$, $SE = 0.16$, $t = 3.90$, $p < .001$ for Japanese confections]. When ratings of expected taste were included as a mediating variable (of the correlation between quality and visit intention ratings), direct effects were not observed for any food type under the label-absent condition [$b = .33$, $SE = 0.16$, $t = 2.04$, $p = .050$ for cheese; $b = .13$, $SE = 0.12$, $t = 1.04$, $p = .305$ for hamburger; $b = .34$, $SE = 0.18$, $t = 1.85$, $p = .074$ for wine; $b = .34$, $SE = 0.18$, $t = 1.84$, $p = .076$ for Japanese confections] or label-present condition [$b = .25$, $SE = 0.14$, $t = 1.78$, $p = .085$ for cheese; $b = .31$, $SE = 0.18$, $t = 1.72$, $p = .096$ for hamburger; $b = .33$, $SE = 0.17$, $t = 1.99$, $p = .056$ for wine; $b = .36$, $SE = 0.22$, $t = 1.63$, $p = .114$ for Japanese confections]. Significant indirect effects were seen for all three food types under the label-absent condition [$b = .26$, $SE = 0.13$, 95% CI [0.060, 0.561] for cheese; $b = .44$, $SE = 0.09$, 95% CI [0.280, 0.622] for hamburger; $b = .23$, $SE = 0.19$, 95% CI [0.031, 0.730] for wine] and label-present condition [$b = .31$, $SE = 0.09$, 95% CI [0.133, 0.498] for cheese; $b = .29$, $SE = 0.12$, 95% CI [0.079, 0.569] for hamburger; $b = .28$, $SE = 0.13$, 95% CI [0.043, 0.572] for wine], except for Japanese confections under both the label-absent [$b = .19$, $SE = 0.11$, 95% CI [−0.173, 0.407]] and label-present conditions [$b = .28$, $SE = 0.18$, 95% CI [−0.007, 0.690].

Overall, the results of Experiment 3 replicated those of Experiments 1 and 2 in that participants provided more positive evaluations of food shops when a past-focused label was present than when the label was absent, except for Japanese confections. The participants' ages (i.e., range: 18–31 years) differed substantially from the ages of the participants in Experiments 1 and 2 (i.e., range: 30–64 years and 28–68 years, respectively). Thus, Experiment 3 ruled out the possibility of an age-related preference (e.g., [18,19]) to explain the congruence effects. Importantly, follow-up univariate tests including food type as a between-subjects factor revealed a large labeling effect only on product quality ratings. Specifically, similar to Experiments 1 and 2, the effect size of the quality ratings ($\eta_p^2 = .137$) was higher than the effect sizes of visit intention ($\eta_p^2 = .041$) and expected taste ($\eta_p^2 = .004$). These results imply that the presentation of a past-focused label modulated participants' evaluations of the food shops regardless of participant age or the evaluated food type.

Experiment 3 showed that the labeling effect for Japanese confections was smaller compared to that observed in Experiments 1 and 2. Importantly, the year-of-establishment labels in the present study involved two types of chronological systems with different cultural meanings, namely the Japanese traditional calendar (Experiments 1 and 2) and the Western (Gregorian) calendar (Experiment 3). These labels were presented to participants to be congruent with the cultural meanings symbolized by the food photographs in the shop advertisements,

with soba noodles accompanied by the Japanese era calendar (Experiments 1 and 2) and foods that originated in the West, except for Japanese confections, accompanied by the Gregorian calendar year label (Experiment 3). This association with regard to cultural meanings is present in advertisements for Japanese cuisine restaurants in Japan. Indeed, our preliminary survey showed that a large number of soba noodle restaurants (77 of 100) advertised their year of establishment using the Japanese era name referring to a medieval or modern period (Fig 1). The types of chronological calendars may impact consumer perception of traditional culture for restaurants because the Japanese era name was the dominant chronological system officially used in Japanese society for thousands of years [28]. Similar impacts of label text type have been demonstrated by a set of marketing studies [10,12] indicating that a text format predominantly used in the culture for extended periods (e.g., texts written vertically in East Asia [29,30]) would be congruent with ratings in terms of past-focused perception. From the theoretical standpoint of the text effects on time-related perception, we propose that Japanese traditional food marketers would benefit from displaying their year of establishment using the corresponding Japanese era name to optimize the effects of temporal congruence on consumers' positive evaluations of their restaurants.

The results of the mediation analysis were virtually identical to those of Experiment 1, where ratings of expected taste modulated the association between product quality evaluations and consumer visit intentions. The analysis consistently indicated significant indirect rather than direct effects of attitudes about quality on visit intentions, irrespective of label condition and food type (i.e., cheese, hamburger, or wine). For Japanese confections, the indirect effects disappeared; intriguingly, the direct effects were also abolished. Despite the absence of direct effects under this condition, the mediation analysis revealed a similar pattern of non-significant mediation effects to that seen under the present-focused-label condition in Experiment 2. We assume that the indirect effects might have been abolished because of the use of present-focused labels or the Gregorian calendar.

Experiments 4 and 5 examined whether displaying a label using the Japanese-era name on the websites of shops serving traditional foods in Japan would optimize temporal congruence effects on consumers' positive evaluations. Accordingly, the present study manipulated label presentations across experiments so that their cultural meanings were consistent or inconsistent with the label's traditional description style. In combination with Japanese traditional food pictures, we displayed the year of establishment of the restaurants using the Gregorian calendar year format in Experiment 4 and in the Japanese era format in Experiment 5. We then required participants to rate their evaluations of the shops. In Experiment 5, the year label was consistent with the shop's picture in terms of schemes of Japanese traditional culture. This consistent association between the label and the shop was expected to enhance the temporal congruence of those materials. Therefore, benefits from the presentation of the past-focused label were presumed to be more robust in Experiment 5 than in Experiment 4. Accordingly, presenting the year of establishment of Japanese traditional shops in the Gregorian calendar year format was expected to reduce the benefit of label presentation because the association with cultural meanings was reduced.

## Experiments 4 and 5

In Experiments 4 and 5, we examined whether the presentation of a label displaying the Japanese era name would be beneficial for advertisements of Japanese traditional foods compared with a label displaying the Gregorian calendar year. We measured participants' evaluations of food shops serving a Japanese classic dessert with a label displaying the Gregorian calendar year (Experiment 4) and with a label displaying the Japanese era name (Experiment 5). Thus,

the cultural meaning of the label's description style in Experiment 5 was consistent with the shops' traditional concept unlike Experiment 4. In Experiments 4 and 5, we compared participants' evaluations between the label-present and label-absent conditions, as in Experiment 1. In accordance with the notion that symbols of written texts would be associated with participants' past-focused perception (e.g., [10]), we predicted that temporal congruence effects would be robust in the presence of a label showing the Japanese era name but not in the presence of a label with the Gregorian calendar year.

## Materials and methods

A new group of 34 participants (9 female participants; mean age = 48.03 years; age range: 23–74 years) for Experiment 4 and another group of 34 participants (14 female participants; mean age = 45.88 years; age range: 21–64 years) for Experiment 5 were recruited from a participant pool at an online crowdsourcing service for survey research.

The Experiments were performed using an online presentation of pictures of fictitious food shop websites (Fig 8) on a web survey system. We used photographs of a Japanese confection (a classic Japanese dessert) as the stimulus, similar to Experiment 3. The pictures of two shops (1,501 × 1,126 pixels) consisted of a main photograph of the food, the name of the shop, and a website navigation menu. A year-of-establishment label (80 × 80 pixels per Japanese character) was embedded horizontally in the space at the top of the main photograph in Experiment 4 or vertically in that space in Experiment 5. In Experiment 4, the year of establishment was displayed as "1912" using the Gregorian calendar year. In contrast, the label in Experiment 5 was displayed in Japanese Taisho 15 era date format ("大正15年"), which corresponds to the year 1926 C.E. Thus, the labels presented to participants in Experiment 5 indicated a time closer to the present than that presented in Experiment 4.

In Experiments 4 and 5, a procedure identical to that used in Experiment 1 was employed, whereby participants observed one of two pictures with the label and the other picture without the label (Fig 8). Participants were instructed to view each picture for 30 seconds and memorize each stimulus. After the presentation of a single picture, participants indicated their evaluations of the confectionary shops in terms of visit intention, expected taste, and product quality using a seven-point scale. At the end of the survey, participants reported a food name presented on the web to test compliance with the task instructions. The data from Experiments 4 and 5 were separately analyzed using a repeated-measures MANOVA, with the label condition (label present vs. label absent) as the within-subjects factor.

## Results and discussion

All participants in both experiments correctly identified the name of the food in the pictures presented on the web survey system. The mean evaluation scores in both experiments were calculated separately for each dependent variable (visit intention, expected taste, and product quality; Fig 9). Repeated-measures MANOVA revealed no significant main effect of the label condition on the ratings in Experiment 4 [$F(3, 31) = 0.34$, $p = .798$, Wilks' $\lambda = .968$, $\eta_p^2 = .032$]. In contrast, there was a significant main effect on the ratings in Experiment 5 [$F(3, 31) = 4.95$, $p = .006$, Wilks' $\lambda = .676$, $\eta_p^2 = .324$]. Follow-up univariate tests with Bonferroni corrections on this main effect revealed a significant effect of label condition on product quality ratings [$F(1, 33) = 13.02$, $p = .003$, $\eta_p^2 = .283$] but no significant effects on ratings of visit intention [$F(1, 33) = 3.16$, $p = .255$, $\eta_p^2 = .087$] and expected taste [$F(1, 33) = 1.98$, $p = .507$, $\eta_p^2 = .057$].

To test whether the attitude toward quality directly or indirectly affected the intention to visit the restaurants, a mediation analysis was performed for each label condition (where 0 = label absent and 1 = label present) and each experiment.

# Experiment 4

創業　1912年　(= 1912 C.E.)

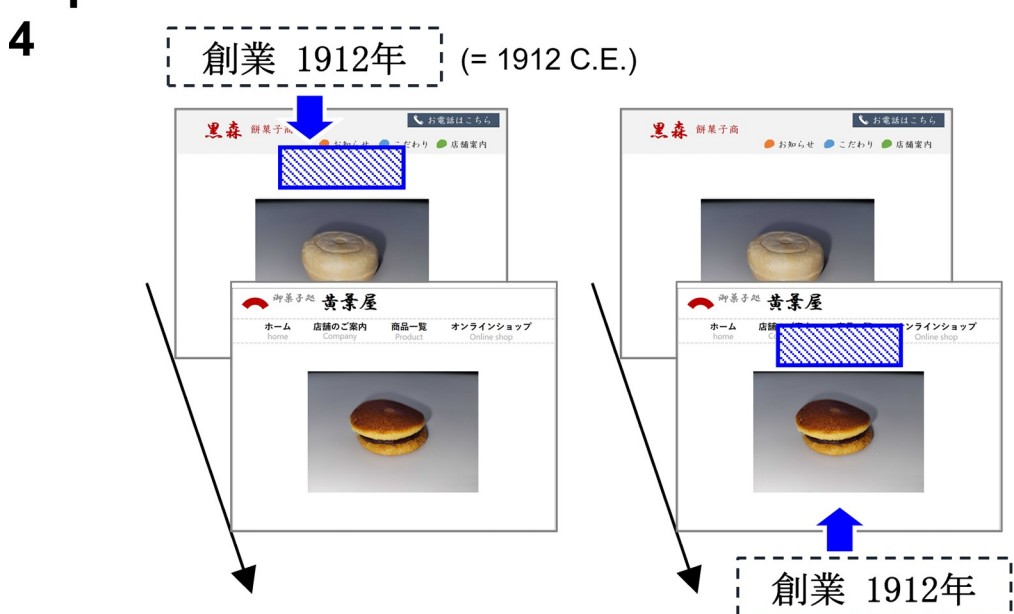

# Experiment 5

創業　大正15年

Taisho 15
(= 1926 C.E.)

創業　大正15年

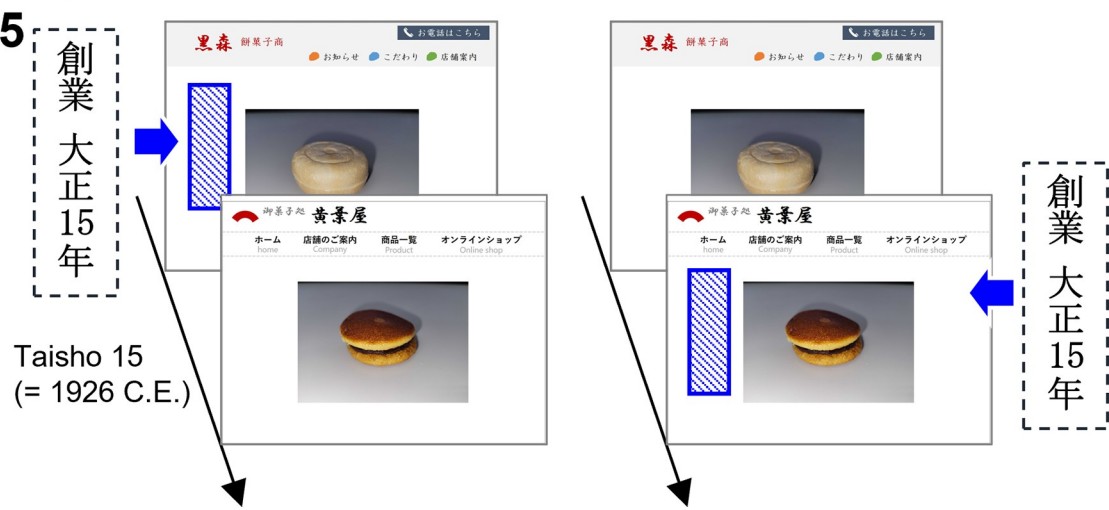

**Fig 8. Schematic of a trial sequence from Experiments 4 and 5.** The restaurant's year of establishment was displayed on the label as 1912 C.E. using the Gregorian calendar in Experiment 4, and as 1926 C.E. using the Taisho era date format in Experiment 5. The label-present and -absent conditions were counterbalanced across participants.

Experiment 4: The results revealed a significant effect of quality ratings on intention ratings under both the label-absent [$b$ = .83, $SE$ = 0.10, $t$ = 8.56, $p <$ .001] and label-present conditions [$b$ = .81, $SE$ = 0.10, $t$ = 8.46, $p <$ .001]. Food taste evaluations directly mediated the correlation between ratings of quality and the intention to visit under the label-absent condition [$b$ = .62, $SE$ = 0.19, $t$ = 3.35, $p$ = .002], but not under the label-present condition [$b$ = .28, $SE$ = 0.16, $t$ = 1.78, $p$ = .085]. The indirect effect was not significant under the label-absent condition [$b$ = .21, $SE$ = 0.28, 95% CI [−0.086, 0.905]] but was significant under the label-present condition [$b$ = .53, $SE$ = 0.10, 95% CI [0.293, 0.702]].

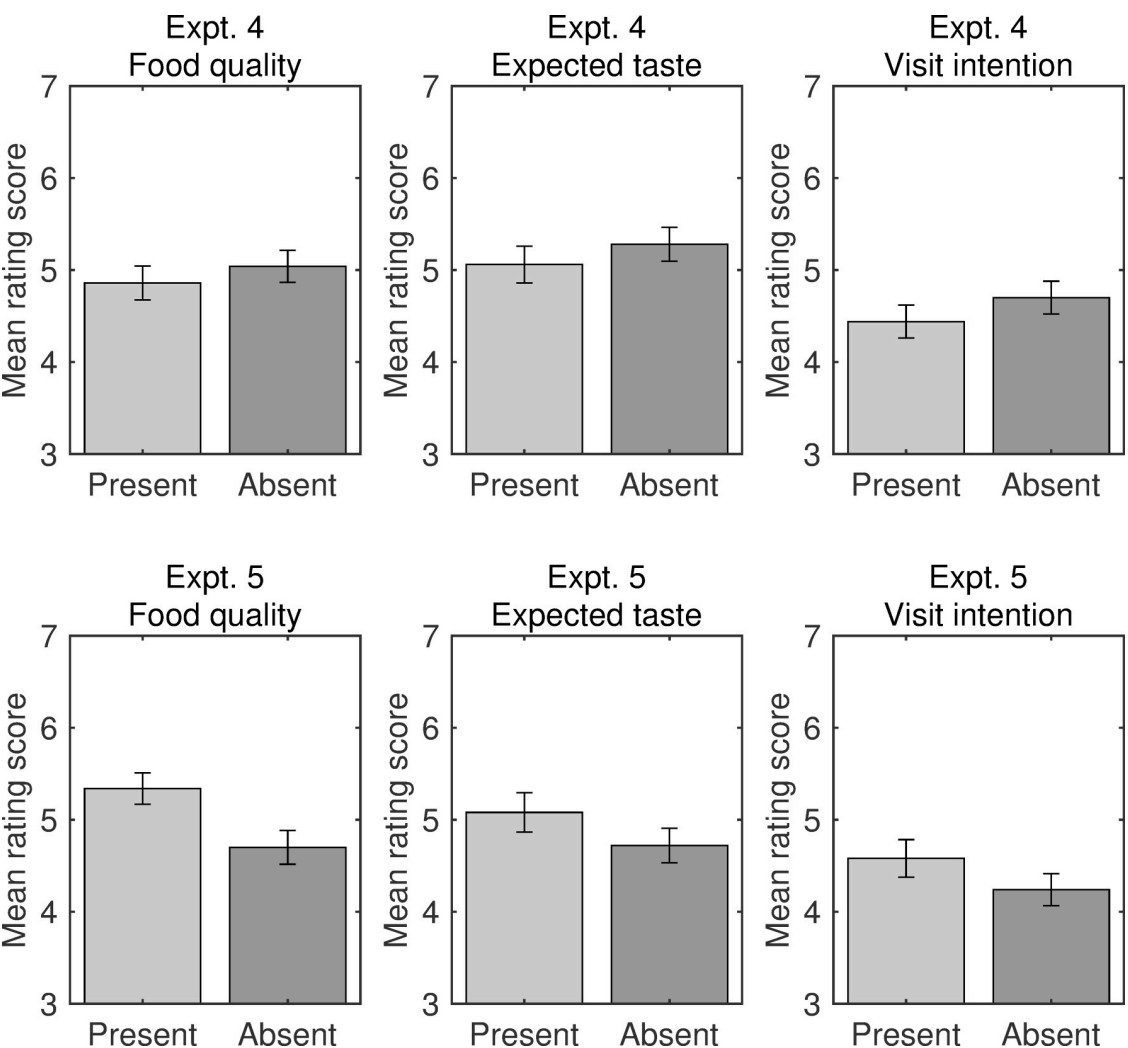

**Fig 9. Evaluations of food shops in Experiments 4 and 5.** Bar plots for visit intention, expected taste, and food quality ratings and calendar type (Gregorian in Experiment 4 and Japanese in Experiment 5) are shown separately for the label present and label absent conditions. Error bars indicate standard error.

Experiment 5: The results revealed a significant effect of quality ratings on intention ratings under the label-absent [$b$ = .45, $SE$ = 0.14, $t$ = 3.18, $p$ = .003] and label-present conditions [$b$ = .80, $SE$ = 0.17, $t$ = 4.70, $p$ < .001]. Food taste evaluations did not directly mediate the correlation between attitudes toward quality and intention to visit under either the label-absent [$b$ = −.06, $SE$ = 0.22, $t$ = −0.28, $p$ = .779] or label-present condition [$b$ = −.03, $SE$ = 0.23, $t$ = −0.12, $p$ = .904]. However, the indirect effect was significant under both the label-absent [$b$ = .51, $SE$ = 0.19, 95% CI [0.111, 0.876]] and label-present conditions [$b$ = .82, $SE$ = 0.27, 95% CI [0.288, 1.350]].

The pattern of results of Experiment 4 contrasted sharply with that in Experiment 5. Experiment 4 revealed that the participants' preference for Japanese traditional food decreased or disappeared in the presence of a label showing the Gregorian calendar year. Experiment 5 showed a preference in the presence of a year label that displayed the Japanese traditional calendar era name. These results suggest that the association between a shop and the label shown, increased the effects of label-presentation in terms of traditional culture on consumers'

positive evaluations of that shop. This congruence effect can be applied to marketing strategies for Japanese traditional restaurants or food shops, with the use of Japanese-era name labels enhancing consumer preferences for those locations.

Consistent with the findings of Experiments 1–3, these preferences were limited in terms of qualitative expectations of the shops shown in Experiment 5. Indeed, the effect size of the quality rating ($\eta_p^2$ = .283) remained higher than the effect sizes of visit intention ($\eta_p^2$ = .087) and expected taste ($\eta_p^2$ = .057). Thus, the results of Experiments 4 and 5 suggest that the Japanese era label symbolized a past era and elicited temporal congruence effects that improved participants' evaluations of the advertisements in the quality domain.

The results of the mediation analysis in Experiment 4 revealed that the indirect effect of attitudes about quality was abolished when the past-focused label was absent. This lack of an indirect effect was consistent with the results of the present-focused label condition, and the results for Japanese confections, in Experiments 2 and 3, implying that the indirect effects were modulated by the type of calendar (Western or Japanese). However, in Experiment 5, in which the labels were presented using the traditional Japanese calendar, the indirect effect was present, consistent with Experiments 1 and 3. Thus, calendar type influences the indirect relationship between ratings of quality and intention to visit.

## General discussion

The present study examined whether the presence of a past-focused label describing an old year of establishment improved positive participant evaluations of advertised shops serving traditional food products. This effect was observed robustly in five experiments in the present study. Experiment 1 indicated that evaluations of the advertised shops were more positive under the label-present than under the label-absent condition. Experiment 2 indicated that evaluations were more positive in the presence of a past-focused label than in the presence of a present-focused label. Experiment 3 replicated the patterns of results found in Experiment 1 using popular Japanese foods other than soba noodles. Experiments 4 and 5 indicated that the impact of label presentation on evaluations of traditional Japanese food shops was strongest when the style of the descriptive label was consistent with the traditional concept of the shop shown in the advertisement. Participants tended to evaluate qualitative aspects of the restaurants and shops more consistently than they evaluated visit intention and expected taste.

In Experiments 1–3, we found that participants evaluated food shops more favorably when a past-focused label was presented than when a present-focused label was presented, or when the label was absent. These patterns can be attributed to the temporal congruence between a past-focused label and a shop with a traditional concept, in accordance with previous reports that materials consistent with conceptual and perceptual orientation aided in relevant information processing [12,15]. However, given that the mere presentation of a label can alter consumers' impressions of food shops or their products in the absence of congruence between the label and to-be-rated material [2–8], it was possible that the patterns of our results might be related to the mere presence of past-focused information, rather than temporal congruence. We ruled out this possibility by performing Experiments 4 and 5. These experiments used two types of temporal labels, the Gregorian calendar year and the Japanese era name. Importantly, the label of the Gregorian calendar year (= 1912) represented a more distant past than did the Japanese era label (= 1926). Accordingly, if past-focused information strongly altered participants' evaluations of the shops, regardless of congruence effects, an improved preference should have been observed in Experiment 4, rather than in Experiment 5. However, participants evaluated the shops favorably when the Japanese era label was present, which was consistent with the shops' concepts of traditional culture. This result emphasizes the importance of

conceptual congruence in enhancing positive evaluations of traditional food shops, as has been demonstrated in previous studies [12,15].

In Experiment 2, we generalized the effects of temporal congruence to popular foods other than soba noodles, such as cheese, hamburgers, and wine. The results indicated that participants had positive evaluations of these food shops with regard to their qualitative aspects when a past-focused label was presented, but not when the label was absent. These results suggest that evaluations of shops improved even when Japanese observers rated shops serving food that originated in the West. Although this topic is not directly relevant to the present study, it should be noted that consumers from non-Japanese cultures (e.g., Western cultures) have been reported to prefer their traditional products [31,32]. Thus, the present results would be applicable to Westerners' shop evaluations for their traditional foods, and possibly for traditional foods in other cultures.

The present study demonstrated consistently positive ratings of food shops in terms of quality across the five experiments. This pattern of results was unique compared with the other ratings. The differences among the three types of ratings is intuitive because they may be associated with different aspects of cognition in the context of assessments of restaurants and shops (see also [12,33,34]). Evaluations of product quality are dissociable from visit intentions, where the former is related to meanings as they pertain to purchase decisions and behaviors. Nevertheless, visit intentions may be based on quality evaluations, and intention ratings predict consumer behavior more accurately than attitude ratings. Moreover, similar to evaluations of product quality, evaluations of expected taste may determine visit intentions, although positive ratings could not be measured via the label condition herein. The results suggested that taste expectations are mainly attributable to information about food products, such as labels denoting organic or low-fat foods [13,22], and sensory information such as texture and mouthfeel [1], rather than to past-focused information. Attitudes toward quality and expected taste are likely among the factors that determine visit intentions, and the labeling effect enhanced the impact of attitudes about quality relative to the other possible determinants of intentions.

Furthermore, differences in the three rating scores might be indirectly highlighted by the notion that past-focused information symbolizes high quality and authenticity (e.g., [17,20]), compared with other ratings. Thus, the present study implies that the applications of label presentation effects may be limited to the enhancement of qualitative evaluations and thus may not be generalizable to evaluations regarding visit intention or expected taste. The greater impact on the quality-focused element of evaluation would not be evoked by an age-rated bias for a traditional brand [18,19] as demonstrated by the present Experiment 3. Participants of all ages may have similar biases related to times and events personally experienced or related to times before the individual was born (see also [35]). Indeed, the results of the present study might be explained by the fact that companies that have been in operation for a number of years are perceived by consumers as producing relatively better (higher quality) products, simply because the company has survived competitive selection over time. For example, if it had produced poor-quality products, the company should have been extinguished from the market. In this case, noting an old year of establishment in an advertisement should signal longevity, and by default, evokes a positive association between the restaurant and its perceived product quality.

In summary, improved quality ratings are sufficient to enhance consumer purchase intentions [36,37]. Using mediation analysis, we found that attitudes to quality were indirectly correlated with visit intentions (via taste expectations). Restaurant advertisements that elicit favorable attitudes toward product quality motivate consumers to visit restaurants due to favorable expectations of taste (Fig 10). This is consistent with the theory of planned behavior.

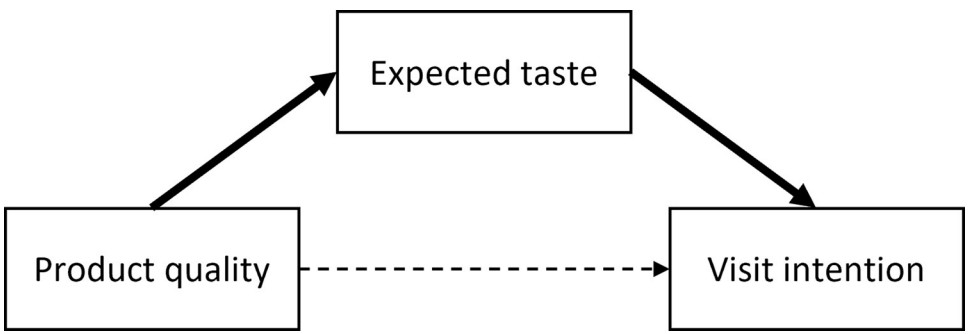

**Fig 10. Relationships among evaluations of product quality, expected taste, and visit intentions.** Although the direct effect of the attitudes toward quality on visit intentions is weak, the indirect effect (through taste expectations) consistently enhances consumer intentions to visit restaurants.

Accordingly, to increase the number of visitors, improving attitudes toward product quality is obviously important. It should be noted that the effects of traditional labeling may be constrained by expectations of restaurants [16]; this may limit the year-of-establishment effect. In [16], advertising increased consumer visit intentions even before they tasted the food products.

As a limitation, this was a small-scale study relative to conventional marketing surveys, although a power analysis was performed. Studies recruiting more consumers from a variety of populations and cultures may provide greater insight into the effects of year-of-establishment labeling.

In summary, the present study found that participants had more favorable evaluations toward food shops and restaurants when web advertisements were accompanied by past-focused labels than when those advertisements were accompanied by a present-focused label or had no such label. The presentation of past-focused information emphasized the shops' quality aspects but did not strongly impact visit intention or expected taste. Moreover, evaluation improvements were more robust when the labels and food shops were consistent in terms of traditional culture, such that the presentation of Japanese traditional food shops was accompanied by a label with a year of establishment displayed using the Japanese era name. These results suggest that temporal congruence between the label and the restaurants or food shops to be rated plays an essential role in ensuring that these advertisements are effective for improving positive evaluations.

## Author Contributions

**Conceptualization:** Tomoki Maezawa, Jun I. Kawahara.

**Formal analysis:** Tomoki Maezawa.

**Investigation:** Tomoki Maezawa.

**Methodology:** Tomoki Maezawa, Jun I. Kawahara.

**Project administration:** Jun I. Kawahara.

**Supervision:** Jun I. Kawahara.

**Writing – original draft:** Tomoki Maezawa.

**Writing – review & editing:** Tomoki Maezawa, Jun I. Kawahara.

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
