## [Decision Letter · Decision Letter 0]

24 Jun 2021

PONE-D-21-15727

A label indicating an old year of establishment improves evaluations of restaurants and shops serving traditional foods

PLOS ONE

Dear Dr. Maezawa,

Thank you for submitting your manuscript to PLOS ONE. After careful consideration, we feel that it has merit but does not fully meet PLOS ONE’s publication criteria as it currently stands. Therefore, we invite you to submit a revised version of the manuscript that addresses the points raised during the review process.

It is this editor opinion that your manuscript can benefit from addressing the two reviewers' comments. I am closer to the first reviewer in what he/she asked to do, so please take particular care on that side.

We look forward to receiving your revised manuscript.

Kind regards,

Luigi Cembalo, PhD

Academic Editor

PLOS ONE

Journal Requirements:

“This work was financially supported by a Grant-in-Aid from the Japan Society for the Promotion of Science Fellows (20J20490) to TM and a Grants-in-Aid for Scientific Research from the Japan Society for the Promotion of Science (20H01779) and (20H04568) to JK. The funders had no role in study design, data collection and analysis, decision to publish, or preparation of the manuscript”

 “This work was financially supported by a Grant-in-Aid from the Japan Society for the Promotion of Science Fellows (20J20490) to TM and a Grants-in-Aid for Scientific Research from the Japan Society for the Promotion of Science (20H01779) and (20H04568) to JK. The funders had no role in study design, data collection and analysis, decision to publish, or preparation of the manuscript.”

“I have read the journal's policy and the authors of this manuscript have the following competing interests”

5. We note that Figures 2, 4 and 6 in your submission contain copyrighted images. All PLOS content is published under the Creative Commons Attribution License (CC BY 4.0), which means that the manuscript, images, and Supporting Information files will be freely available online, and any third party is permitted to access, download, copy, distribute, and use these materials in any way, even commercially, with proper attribution. For more information, see our copyright guidelines: http://journals.plos.org/plosone/s/licenses-and-copyright.

 a. You may seek permission from the original copyright holder of Figures 2, 4 and 6 to publish the content specifically under the CC BY 4.0 license.

Reviewers' comments:

Reviewer's Responses to Questions

**Comments to the Author**

1. Is the manuscript technically sound, and do the data support the conclusions?

Reviewer #1: Yes

Reviewer #2: Yes

2. Has the statistical analysis been performed appropriately and rigorously? 

Reviewer #1: Yes

Reviewer #2: Yes

3. Have the authors made all data underlying the findings in their manuscript fully available?

Reviewer #1: Yes

Reviewer #2: Yes

4. Is the manuscript presented in an intelligible fashion and written in standard English?

Reviewer #1: Yes

Reviewer #2: Yes

5. Review Comments to the Author

Reviewer #1: The paper entitled "A label indicating an old year of establishment improves evaluations of restaurants and shops serving traditional foods" describes a research that aims to evaluate the effect of exposure to past-focused labels (i.e., year of establishment) on attitudes and intention to visit restaurants and shops serving traditional foods in Japan. On the whole, the paper and the experiments presented therein are interesting and methodologically detailed. I report my specific suggestions to improve the paper below.

Introduction

In several psychological theories and models (e.g., Theory of Planned Behaviour) the intention to take action, like visiting a restaurant, is not on the same level of attitudes. In socio-cognitive models of behavioural change attitudes predict intentions which in turn predict behaviours. The Authors could refer specifically to these theoretical aspects in the introduction.

Page 6. Lines 101-103. In line with the above, the Authors should reformulate the sentence presenting the three considered dependent variables with a brief theoretical explanation.

Minor comment

Page 3. Lines 55-58. “Specifically, displaying an old year of establishment improves qualitative aspects of products more positively than other impressions that the time-related labels do not symbolize”. This sentence is not very clear. Please, reformulate.

Page 5. Line 90. Just a typo: there is both the definite and the indefinite article.

Experiment 1

Page 9. Lines 157-159. Following what I suggested in the previous comment, I would write something like so: “… participants indicated their attitudes toward the restaurant (expected taste and product quality) and their intention to visit on a seven-point scale…”. The authors should better describe how they measured attitudes and all the other variables used in the experiment (for the other experiments as well).

Page 9. Line 162. The Authors should state which software was used for statistical analysis (for the other experiments as well).

Page 9. Line 165. I wonder why for the first experiment there are two different sections for results and discussion while for the other experiments results and discussion are presented together. If there is no specific reason for that, please, just for the sake of clarity, choose one way or the other.

The article presents 5 experiments, described in detail. To facilitate the reader,

my advice is to create a summary table of the 5 experiments (something similar to the table below)

Experiment Subjects Experimental Conditions Measures Analyses Summary of results

1

2

3

4

5

General Discussion

Pages 27-32. Following the theoretical differences between attitudes and intentions, please correct all the sentences in which visit intention is mentioned as an attitude.

Page 30. Lines 543-546. This part about the “why” and the “how” is not very clear. Please, be more specific.

Reviewer #2: The manuscript is interesting and adds useful information to the literature. The five experiments carried out are well explained, and the writing (according to my expertise) is formal and fluent. The results and conclusions are interesting.

However, I would recommend adding some elements

1) Immediately after the introduction, you could insert a short section in which you elaborate on the theoretical basis on which the study was based (you could elaborate on the consumer theories that gave you the inspiration to develop the study, the type of experiments or the context in which the study is developed and you could emphasise its originality).

2) In the final part of the conclusions you could include the limitations of the study (e.g. is it a convenience sample? is it large enough and heterogeneous enough? does it analyse a large area of Japan?) and the implications of the results (theoretical, political and managerial implications).

Finally, I think that in line 48 the reference is missing (after the word "product"), and in line 90 you should delete one of the articles between "the" and "a".

Points 1 and 2 are suggestions, but I think they can add value to your study.

6. PLOS authors have the option to publish the peer review history of their article (what does this mean?). If published, this will include your full peer review and any attached files.

Reviewer #1: No

Reviewer #2: No

---

## [Author Response · Author response to Decision Letter 0]

24 Jul 2021

>It is this editor opinion that your manuscript can benefit from addressing the two reviewers' comments. I am closer to the first reviewer in what he/she asked to do, so please take particular care on that side.

Thank you for your decision regarding our manuscript (PONE-D-21-15727). We have revised the main text and figures to meet the journal requirements and have responded to the two reviewers’ comments. 

--------------

>Journal Requirements:

>[1]. Please ensure that your manuscript meets PLOS ONE's style requirements, including those for file naming. The PLOS ONE style templates can be found at …

We have revised the style templates according to the requirements of PLOS ONE, for both the title page and main body of the manuscript. 

>[2]. We note that you have provided funding information that is not currently declared in your Funding Statement. However, funding information should not appear in the Acknowledgments section or other areas of your manuscript. We will only publish funding information present in the Funding Statement section of the online submission form.

>Please include your amended statements within your cover letter; we will change the online submission form on your behalf.

We have moved the funding information from the Acknowledgments section to the cover letter.

>[3]. Please complete your Competing Interests on the online submission form to state any Competing Interests. If you have no competing interests, please state "The authors have declared that no competing interests exist.", as detailed online in our guide for authors at http://journals.plos.org/plosone/s/submit-now

>This information should be included in your cover letter; we will change the online submission form on your behalf.

We have no competing interests, which is now declared in the cover letter.

>[4]. In your Data Availability statement, you have not specified where the minimal data set underlying the results described in your manuscript can be found. PLOS defines a study's minimal data set as the underlying data used to reach the conclusions drawn in the manuscript and any additional data required to replicate the reported study findings in their entirety. All PLOS journals require that the minimal data set be made fully available. For more information about our data policy, please see http://journals.plos.org/plosone/s/data-availability.

>Upon re-submitting your revised manuscript, please upload your study’s minimal underlying data set as either Supporting Information files or to a stable, public repository and include the relevant URLs, DOIs, or accession numbers within your revised cover letter. For a list of acceptable repositories, please see http://journals.plos.org/plosone/s/data-availability#loc-recommended-repositories. Any potentially identifying patient information must be fully anonymized.

 The data set and stimuli will be available through the Open Science Framework repository (https://osf.io/nf8sw/). A statement to this effect has been added to the cover letter.

>[5]. We note that Figures 2, 4 and 6 in your submission contain copyrighted images. All PLOS content is published under the Creative Commons Attribution License (CC BY 4.0), which means that the manuscript, images, and Supporting Information files will be freely available online, and any third party is permitted to access, download, copy, distribute, and use these materials in any way, even commercially, with proper attribution. For more information, see our copyright guidelines: http://journals.plos.org/plosone/s/licenses-and-copyright.

>We require you to either (1) present written permission from the copyright holder to publish these figures specifically under the CC BY 4.0 license, or (2) remove the figures from your submission:

>If you are unable to obtain permission from the original copyright holder to publish these figures under the CC BY 4.0 license or if the copyright holder’s requirements are incompatible with the CC BY 4.0 license, please either i) remove the figure or ii) supply a replacement figure that complies with the CC BY 4.0 license. Please check copyright information on all replacement figures and update the figure caption with source information. If applicable, please specify in the. figure caption text when a figure is similar but not identical to the original image and is therefore for illustrative purposes only.

 The food pictures used in Figure 2 were taken in our laboratory, and we have replaced the copyrighted pictures with non-copyrighted ones (taken in our laboratory) in Figures 4 and 6. We have added the following text to the caption of Figure 4: “Food pictures are similar, but not identical, to the original images and are for illustrative purposes only.”

--------------

>Reviewer #1 Major comment

>[1] Introduction- In several psychological theories and models (e.g., Theory of Planned Behaviour) the intention to take action, like visiting a restaurant, is not on the same level of attitudes. In socio-cognitive models of behavioural change attitudes predict intentions which in turn predict behaviours. The Authors could refer specifically to these theoretical aspects in the introduction.

 Thank you for your helpful comment. As suggested, we have emphasized the relationships among product quality, expected taste, and visit intention in the revised version of manuscript. We introduced a mediation model (see Figure 10), in which visit intention is dissociable from the other two attitudes. In this model, quality ratings predict taste expectations. Accordingly, the regression model included expected taste as a mediator, quality rating as an independent variable, and visit intention as a dependent variable.

 We performed mediation analysis and MANOVA separately for each food type (pages 27–28, lines 418–439) in Experiment 3. The results and conclusions of the mediation analysis have been included in the manuscript (page 13, lines 195–208 and pages 16–17, lines 226–233 for Experiment 1; pages 20–21, lines 300–312 and pages 22–23, lines 324–334 for Experiment 2; pages 29–30, lines 451–473 and pages 32–33, lines 511–521 for Experiment 3; pages 38–39, lines 608–627 and page 40, lines 645–653 for Experiments 4 and 5; and page 45, lines 736–750 for the General Discussion). 

 Because we primarily focused on evaluations of product quality, we have modified the order of presentation of the dependent variables in Figures 3, 5, 7, and 9, such that product quality is followed by expected taste and visit intention.

>[2] Page 6. Lines 101-103. In line with the above, the Authors should reformulate the sentence presenting the three considered dependent variables with a brief theoretical explanation.

 According to this comment, we have explained the meanings of and relationships among the three dependent variables in the Introduction (pages 6–7, lines 106–117). We also replaced the term “attitude” with “evaluation.”

>Minor comment

>[3] Page 3. Lines 55-58. “Specifically, displaying an old year of establishment improves qualitative aspects of products more positively than other impressions that the time-related labels do not symbolize”. This sentence is not very clear. Please, reformulate.

 We have revised the sentence (pages 5–6, lines 84–90) as follows: “Specifically, providing past-focused information, such as the year of establishment, promotes more favorable product appraisals. This effect is similar to the label-presentation effects seen for well-known brands [1, 21] and traditional production processes [16] emphasizing historic aspects. Based on the idea that these associations are considered to indicate quality, we assumed that the presentation of the year of establishment in advertisements could improve consumer attitudes toward restaurants.”

>[4]Page 5. Line 90. Just a typo: there is both the definite and the indefinite article.

 This typo has been corrected.

>[5] Experiment 1- Page 9. Lines 157-159. Following what I suggested in the previous comment, I would write something like so: “… participants indicated their attitudes toward the restaurant (expected taste and product quality) and their intention to visit on a seven-point scale…”. The authors should better describe how they measured attitudes and all the other variables used in the experiment (for the other experiments as well).

 We have replaced the term “attitude” with “evaluation.” The participants were required to answer two attitudinal questions (“How good is the quality?” and “How good is the taste?”) and one intention-related question (“How badly do you want to visit?”) on a seven-point scale (pages 11–12, lines 167–177). The procedure used in Experiments 2–5 was identical to that used in Experiment 1.

>[6] Page 9. Line 162. The Authors should state which software was used for statistical analysis (for the other experiments as well).

 We used SPSS software (page 12, line 177). The mediation analysis was performed using Model 4 of Hayes’ PROCESS SPSS macros (page 13, lines 196).

>[7] Page 9. Line 165. I wonder why for the first experiment there are two different sections for results and discussion while for the other experiments results and discussion are presented together. If there is no specific reason for that, please, just for the sake of clarity, choose one way or the other.

 As suggested, we combined the Discussion and Results sections.

>[8] The article presents 5 experiments, described in detail. To facilitate the reader,my advice is to create a summary table of the 5 experiments (something similar to the table below) >Experiment Subjects Experimental Conditions Measures Analyses Summary of results

 We created a table (Table 1) that summarizes the experimental conditions. We also added Table 2, which briefly summarizes the MANOVA and mediation analysis results. 

>[9] General Discussion- Pages 27-32. Following the theoretical differences between attitudes and intentions, please correct all the sentences in which visit intention is mentioned as an attitude.

 We have replaced the term “attitude” with “evaluation.”

>[10] Page 30. Lines 543-546. This part about the “why” and the “how” is not very clear. Please, be more specific.

 This sentence has been revised as follows: “The present study demonstrated consistently positive ratings of food shops in terms of quality across the five experiments. This pattern of results was unique compared with the other ratings. The differences among the three types of ratings is intuitive because they may be associated with different aspects of cognition in the context of assessments of restaurants and shops (see also [12, 33, 34]). Evaluations of product quality are dissociable from visit intentions, where the former is related to meanings as they pertain to purchase decisions and behaviors. Nevertheless, visit intentions may be based on quality evaluations, and intention ratings predict consumer behavior more accurately than attitude ratings. Moreover, similar to evaluations of product quality, evaluations of expected taste may determine visit intentions, although positive ratings could not be measured via the label condition herein. The results suggested that taste expectations are mainly attributable to information about food products, such as labels denoting organic or low-fat foods [13, 22], and sensory information such as texture and mouthfeel [1], rather than to past-focused information. Attitudes toward quality and expected taste are likely among the factors that determine visit intentions, and the labeling effect enhanced the impact of attitudes about quality relative to the other possible determinants of intentions.” (pages 43–44, lines 702–718).

--------------

>Reviewer #2 Major comment

>[1] Immediately after the introduction, you could insert a short section in which you elaborate on the theoretical basis on which the study was based (you could elaborate on the consumer theories that gave you the inspiration to develop the study, the type of experiments or the context in which the study is developed and you could emphasise its originality).

 We have embellished the discussion of the theoretical basis of the present study in the Introduction section (page 6, lines 91–98). We focused on whether the effects of labeling emphasizing traditional aspects improved attitudes toward product quality. The benefits of labeling can be explained by temporal congruence and an emphasis on tradition.

>[2] In the final part of the conclusions you could include the limitations of the study (e.g. is it a convenience sample? is it large enough and heterogeneous enough? does it analyse a large area of Japan?) and the implications of the results (theoretical, political and managerial implications).

 We now discuss the limitations of the present study in the General Discussion section (page 45, lines 742–745 and page 46, lines 752–755). 

>[3] Finally, I think that in line 48 the reference is missing (after the word "product"), and in line 90 you should delete one of the articles between "the" and "a".

 We have added the reference to the text and corrected the typo.

Thank you for your encouraging and helpful comments. We look forward to your editorial decision.

Sincerely,

Tomoki Maezawa

Jun I. Kawahara

---

## [Decision Letter · Decision Letter 1]

12 Oct 2021

A label indicating an old year of establishment improves evaluations of restaurants and shops serving traditional foods

PONE-D-21-15727R1

Dear Dr. Maezawa,

We’re pleased to inform you that your manuscript has been judged scientifically suitable for publication and will be formally accepted for publication once it meets all outstanding technical requirements.

Kind regards,

Luigi Cembalo, PhD

Academic Editor

PLOS ONE

Additional Editor Comments (optional):

Reviewers' comments:

Reviewer's Responses to Questions

**Comments to the Author**

1. If the authors have adequately addressed your comments raised in a previous round of review and you feel that this manuscript is now acceptable for publication, you may indicate that here to bypass the “Comments to the Author” section, enter your conflict of interest statement in the “Confidential to Editor” section, and submit your "Accept" recommendation.

Reviewer #1: (No Response)

Reviewer #2: All comments have been addressed

2. Is the manuscript technically sound, and do the data support the conclusions?

Reviewer #1: (No Response)

Reviewer #2: Yes

3. Has the statistical analysis been performed appropriately and rigorously? 

Reviewer #1: (No Response)

Reviewer #2: Yes

4. Have the authors made all data underlying the findings in their manuscript fully available?

Reviewer #1: (No Response)

Reviewer #2: Yes

5. Is the manuscript presented in an intelligible fashion and written in standard English?

Reviewer #1: (No Response)

Reviewer #2: Yes

6. Review Comments to the Author

Reviewer #1: The authors replied to all comments and improved the article (in particular they better defined the theoretical aspects, clarifying the difference between attitude and behavior)

Reviewer #2: I re-read the paper and saw that the authors have implemented all the corrections previously suggested by the various reviewers. The implementations were comprehensive and filled the existing gaps. The topic is interesting and now all the sections are well discussed and properly explored. I think the paper can be considered completed.

7. PLOS authors have the option to publish the peer review history of their article (what does this mean?). If published, this will include your full peer review and any attached files.

Reviewer #1: No

Reviewer #2: No

---

## [Editor Report · Acceptance letter]

15 Oct 2021

PONE-D-21-15727R1 

A label indicating an old year of establishment improves evaluations of restaurants and shops serving traditional foods 

Dear Dr. Maezawa:

I'm pleased to inform you that your manuscript has been deemed suitable for publication in PLOS ONE. Congratulations! Your manuscript is now with our production department. 

Kind regards, 

on behalf of

Dr. Luigi Cembalo 

Academic Editor

PLOS ONE